# HIERARCHY-AWARE BIASED BOUND LOSS FUNCTION FOR HIERARCHICAL TEXT CLASSIFICATION

## ABSTRACT

Hierarchical text classification (HTC) is a challenging problem with two key issues: utilizing structural information and mitigating label imbalance. Recently, the unit-based approach generating unit-based feature representations has outperformed the global approach focusing on a global feature representation. Nevertheless, unit-based models using BCE and ZMLCE losses still face static thresholding and label imbalance challenges. Those challenges become more critical in large-scale hierarchies. This paper introduces a novel hierarchy-aware loss function for unit-based HTC models: Hierarchy-aware Biased Bound (H2B) loss. H2B integrates learnable bounds and biases to address static thresholding and mitigate label imbalance adaptively. Experimental results on benchmark datasets demonstrate the superior performance of H2B compared to competitive HTC models.

## 1 INTRODUCTION

Hierarchical Text Classification (HTC) aims to classify text into a predefined label hierarchy. HTC currently faces two fundamental challenges: utilizing structural information and mitigating label imbalance. As shown in Figure 1, recent research can be categorized into global and unit-based approaches based on exploiting feature representations combined with text and structural information. The global approach, HiAGM (Zhou et al., 2020), HiMatch (Chen et al., 2021), HGCLR (Wang et al., 2022a), K-HTC (Liu et al., 2023), and HiTIN (Zhu et al., 2023), generates a holistic feature representation of text that encompasses an entire hierarchy and use it to compute label scores comprehensively. In contrast, the unit-based approach, HPT (Wang et al., 2022b) and HiDEC (Im et al., 2023), generate unit-level feature representations of text, where a unit refers to a subset of a hierarchy partitioned according to specific strategies, and classification is performed on labels within these units. The unit-based approach has achieved significant improvements over the global approach.

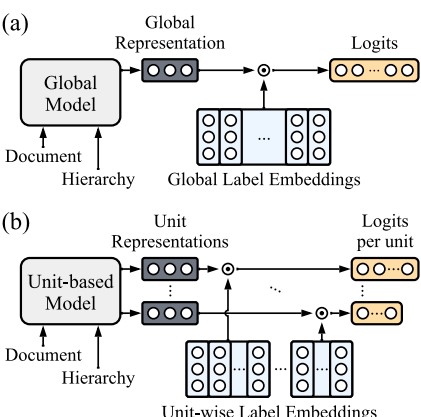

Figure 1: Classification processes of (a) Global and (b) Unit-based HTC models.

However, two significant limitations have emerged from existing research: static thresholding and label imbalance. Static thresholding is problematic because most HTC models using binary cross entropy (BCE) predict positive labels using a fixed threshold, typically set at 0.5, when the output probability exceeds this threshold. Determining optimal thresholds for target labels is challenging as it requires extensive computations for various units. Label imbalance is also a concern because frequent labels dominate training data, resulting in numerous false negatives. Unit-based HTC models, such as HPT and HiDEC, address this issue by rebalancing the ratio between positives and negatives by sampling a subset of a hierarchy as units. In particular, HPT presents a zero-bounded multi-label cross entropy (ZMLCE) loss using a log-sum-exp operation to deal with the imbalance, but the static threshold problem still remains.

To tackle these limitations, this paper introduces a novel hierarchy-aware loss function for unit-based HTC models: Hierarchy-aware Biased Bound (H2B) loss. Our key innovations in H2B are as follows. First, we incorporate learnable bounds for all units within a hierarchy to address static thresholding. These bounds are optimized for various units during training and are used as dynamic thresholds during inference. Second, in addition to the bounds, we introduce biases for both positive and negative labels to alleviate the imbalance resulting from overtraining dominant negative labels. Biases are only used to measure the degree of label imbalance within units during training and are not utilized during inference. A bias for a positive label amplifies the corresponding bound, whereas one for a negative label diminishes the corresponding bound. Consequently, the biases help to adjust the bounds to reduce the impact of the ratio between positive and negative labels.

Through a series of experiments, we demonstrate the effectiveness of our loss function applied to recent unit-based HTC models, HPT and HiDEC, using three benchmark datasets: RCV1-v2 (Lewis et al., 2004), NYT (Sandhaus, 2008), and EURLEX57K (Chalkidis et al., 2019). Notably, our loss function outperforms competitive HTC models on all three benchmark datasets. We comprehensively analyze how the bounds and biases influence static thresholding and label imbalance in HTC.

Our contributions are summarized as follows:

- We propose a novel hierarchy-aware loss function, H2B, for unit-based HTC models to address static threshold and label imbalance by incorporating learnable bounds and biases. The bounds are optimized during training and used as dynamic thresholds during inference, with biases helping adjust the bounds to reduce the impact of the ratio between positive and negative labels.

- We demonstrate the effectiveness of our loss function applying to recent unit-based HTC models by comparing competitive HTC models on three benchmark datasets. Our results confirm the superiority and behaviors of our loss function, supported by in-depth analysis.

## 2 RELATED WORK

Recent HTC research based on deep learning can be categorized into global and unit-based approaches, each with its unique way of creating feature representations that incorporate both text and hierarchy structure.

The unit-based approach generates feature representations at the unit-level by partitioning the entire hierarchy into units using specific strategies. Each unit corresponds to a subset of labels within a hierarchy. Various models employ diverse unit construction strategies, including "for-each-class" (Banerjee et al., 2019), "for-each-parent" (Kowsari et al., 2017; Im et al., 2023), "for-each-level" (Shimura et al., 2018; Wang et al., 2022b), and "for-each-sub-hierarchy" (Peng et al., 2018). HDL-Tex (Kowsari et al., 2017) introduces HTC models using DNN, CNN, and RNN architectures. HTrans (Banerjee et al., 2019) enhances HDLTex by employing transfer learning to preserve path information. HR-DGCNN (Peng et al., 2018) utilizes recursive hierarchical segmentation to divide a hierarchy into sub-hierarchies and construct local unit models. However, the unit-based approach often suffers from a lack of hierarchical information.

In contrast, the global approach generates a holistic feature representation encompassing the entire label hierarchy. HiAGM (Zhou et al., 2020) merges text and structural representations through text propagation, while HGCLR (Wang et al., 2022a) propagates structural representation through a text encoder and employs contrastive learning. HiMatch (Chen et al., 2021) applies a hierarchy-aware matching loss to HiAGM and adjusts feature representations based on hierarchy information. K-HTC (Liu et al., 2023) tries to incorporate a knowledge graph into HTC using knowledge-aware hierarchical label attention and contrastive learning. HiTIN (Zhu et al., 2023) reduces the complexity of the existing global models by reconstructing a hierarchy to minimize structural entropy. The global models effectively leverage hierarchical information through structure encoders (Kipf & Welling, 2017; Ying et al., 2021), outperforming unit-based models. Despite their achievements, they face challenges of label imbalances and hierarchy-dependent model parameters.

To address these challenges, HPT (Wang et al., 2022b) and HiDEC (Im et al., 2023) incorporate a structure encoder (Veličković et al., 2018) and attention mechanism (Vaswani et al., 2017) into their unit-based HTC models. HiDEC utilizes an encoder-decoder architecture to generate a sub-

hierarchy sequence based on the target labels of each document using a parent-level unit construction strategy. By dividing a hierarchy based on levels, HPT integrates level-specific feature representations from a structure encoder into a text encoder and proceeds with unit-wise prediction. Furthermore, HPT introduces ZMLCE loss (Su et al., 2022) to deal with label imbalance by adding a zero-bound to the existing MLCE loss (Sun et al., 2020; Li et al., 2017). However, these methods still encounter label imbalance in large-scale hierarchies and suffer from static thresholding.

# 3 PROPOSED HIERARCHY-AWARE LOSS FUNCTION

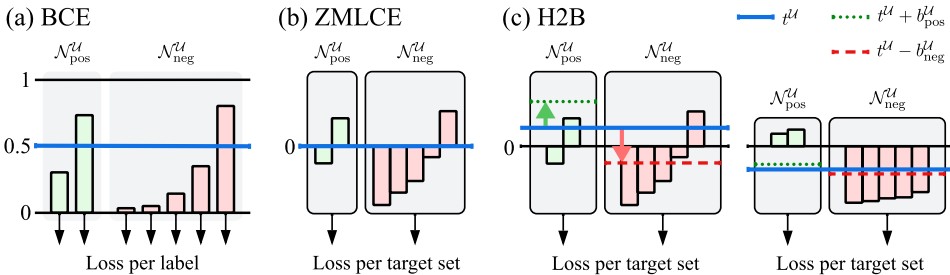

Figure 2: The illustration of classification losses, (a) BCE, (b) ZMLCE, and (c) H2B. The blue line is a threshold during inference. In BCE, a loss is computed for each label and then averaged. In ZMLCE and H2B, a loss is calculated separately for positive and negative target sets and combined. In H2B, the bound is optimized for each unit and used as dynamic thresholds during inference. The green and red lines are positive and negative biased bounds, respectively, during training.

## 3.1 PRELIMINARIES AND NOTATIONS

Let a graph $\mathcal{G} = (\mathcal{V}, \mathcal{E})$ be a predefined hierarchy where $\mathcal{V} = \{v_1, \dots, v_N\}$ is a set of all label nodes and $\mathcal{E} = \{(v_i, v_j) | v_i, v_j \in \mathcal{V}\}$ is a set of edges indicating a relation between two nodes. $\mathcal{D} = \{(x_d, \mathcal{Y}_d)\}_{d=1}^{|D|}$ is a document dataset where $x_d$ is $d$-th document and $\mathcal{Y}_d \subset \mathcal{V}$ is a set of target labels associated with $x_d$. Note $|\mathcal{Y}_d| \geq 1$ because a document $x_d$ can have multi-labels. We partition $\mathcal{V}$ into a set of units $\mathcal{W} = \{\mathcal{U}_1, \dots, \mathcal{U}_{|\mathcal{W}|}\}$ where $\mathcal{U}$ denotes a unit composed of a set of labels.

For a given document $x_d$, unit-based HTC models generate a unit representation $r^{\mathcal{U}}$, then compute logits $l^{\mathcal{U}}$ using the unit representation $r^{\mathcal{U}}$ and label embeddings associated with the labels in a unit $\mathcal{U}$. These logits $l^{\mathcal{U}}$ are employed to make predictions on a unit $\mathcal{U}$. The target label set for each unit is defined as $\mathcal{Y}_d^{\mathcal{U}} = \{v_i | v_i \in (\mathcal{Y}_d \cap \mathcal{U})\}$.

To calculate a loss, we divide a unit $\mathcal{U}$ into positive and negative target sets, denoted as $\mathcal{N}_{\text{pos}}^{\mathcal{U}} = \{v_i | v_i \in \mathcal{Y}_d^{\mathcal{U}}\}$ and $\mathcal{N}_{\text{neg}}^{\mathcal{U}} = \{v_i | v_i \in \mathcal{U} \backslash \mathcal{Y}_d^{\mathcal{U}}\}$. If the target label does not exist within a specific unit, $\mathcal{N}_{\text{pos}}^{\mathcal{U}}$ can become an empty set. Based on $\mathcal{N}_{\text{pos}}^{\mathcal{U}}$ and $\mathcal{N}_{\text{neg}}^{\mathcal{U}}$, in Figure 2-(a), BCE loss is defined as:

$$\mathcal{L}_{\text{BCE}} = -\frac{1}{\sum_{\mathcal{U} \in \mathcal{W}} |\mathcal{U}|} \sum_{\mathcal{U} \in \mathcal{W}} \left( \sum_{p \in \mathcal{N}_{\text{pos}}^{\mathcal{U}}} \log \sigma \left( l_p^{\mathcal{U}} \right) + \sum_{n \in \mathcal{N}_{\text{neg}}^{\mathcal{U}}} \log \left( 1 - \sigma \left( l_n^{\mathcal{U}} \right) \right) \right) \tag{1}$$

where $l_p^{\mathcal{U}}$ and $l_n^{\mathcal{U}}$ are the logits for positive label $p$ and negative label $n$, respectively. $\sigma(\cdot)$ is a sigmoid function.

BCE loss has a weakness in dealing with label imbalance. To this end, ZMLCE (Zero-bounded Multi-Label Cross-Entropy) loss is presented in HPT (Wang et al., 2022b) :

$$\mathcal{L}_{\text{ZMLCE}} = \frac{1}{|\mathcal{W}|} \sum_{\mathcal{U} \in \mathcal{W}} \left( \log \left( 1 + \sum_{p \in \mathcal{N}_{\text{pos}}^{\mathcal{U}}} e^{-l_p^{\mathcal{U}}} \right) + \log \left( 1 + \sum_{n \in \mathcal{N}_{\text{neg}}^{\mathcal{U}}} e^{l_n^{\mathcal{U}}} \right) \right). \tag{2}$$

As depicted in Figure 2-(b), while ZMLCE loss attempts to mitigate label imbalance through the log-sum-exp operation, it does not address static thresholding because the bounds for all units remain fixed at 0.

## 3.2 HIERARCHY-AWARE BIASED BOUND LOSS

We propose a Hierarchy-aware Biased Bound (H2B) loss to simultaneously address the issues of static thresholding and label imbalance within a unit $\mathcal{U}$. H2B is defined as:

$$\mathcal{L}_{\text{H2B}} = \frac{1}{|\mathcal{W}|} \sum_{\mathcal{U} \in \mathcal{W}} \left( \log \left( 1 + \sum_{p \in \mathcal{N}_{\text{pos}}^{\mathcal{U}}} e^{-l_p^{\mathcal{U}} + \left(t^{\mathcal{U}} + b_{\text{pos}}^{\mathcal{U}}\right)} \right) + \log \left( 1 + \sum_{n \in \mathcal{N}_{\text{neg}}^{\mathcal{U}}} e^{l_n^{\mathcal{U}} - \left(t^{\mathcal{U}} - b_{\text{neg}}^{\mathcal{U}}\right)} \right) \right) \quad (3)$$

where $t^{\mathcal{U}} \in \mathbb{R}$ is a learnable bound for a unit $\mathcal{U}$. $b_{\text{pos}}^{\mathcal{U}}$ and $b_{\text{neg}}^{\mathcal{U}}$ are positive and negative biases for a unit $\mathcal{U}$, respectively.

The bound $t^{\mathcal{U}}$ is computed using a unit representation $r^{\mathcal{U}}$, allowing us to predict distinct bounds for each unit by employing text and hierarchy information. The biases $b_{\text{pos}}^{\mathcal{U}}$ and $b_{\text{neg}}^{\mathcal{U}}$ can be computed using any function $g : \mathcal{N} \to \mathbb{R}^{+}$ designed to mitigate label imbalance on $\mathcal{N}_{\text{pos}}^{\mathcal{U}}$ and $\mathcal{N}_{\text{neg}}^{\mathcal{U}}$. We employ a standard deviation, $g = \text{std}(\{l_{v_i}^{\mathcal{U}} | v_i \in \mathcal{N}\})$. Like Figure 2-(c), a high standard deviation of logits indicates insufficient model training on the labels within $\mathcal{N}$, leading to the assignment of higher biases. This adjustment increases the corresponding bound, providing a better opportunity to train on less frequent labels as the bound is elevated. The bound $t^{\mathcal{U}}$ is optimized and used as dynamic thresholds by $\hat{\mathcal{Y}}_d^{\mathcal{U}} = \{v_i | l_{v_i}^{\mathcal{U}} > t^{\mathcal{U}}, v_i \in \mathcal{U}\}$ during inference. The biases $b_{\text{pos}}^{\mathcal{U}}$ and $b_{\text{neg}}^{\mathcal{U}}$ are only used with detaching gradients during training.

**Comparison to ZMLCE Loss**  For a deeper understanding, we compare H2B loss with ZMLCE loss. Equation 4 is derived from ZMLCE loss in Equation 2.

$$\mathcal{L}_{\text{ZMLCE}}^{\mathcal{U}} = \log(1 + \underbrace{\sum_{p \in \mathcal{N}_{\text{pos}}^{\mathcal{U}}} \sum_{n \in \mathcal{N}_{\text{neg}}^{\mathcal{U}}} e^{l_n^{\mathcal{U}} - l_p^{\mathcal{U}}}}_{(a)} + \underbrace{\sum_{p \in \mathcal{N}_{\text{pos}}^{\mathcal{U}}} e^{0 - l_p^{\mathcal{U}}} + \sum_{n \in \mathcal{N}_{\text{neg}}^{\mathcal{U}}} e^{l_n^{\mathcal{U}} - 0}}_{(b)}) \quad (4)$$

ZMLCE loss can be split into two terms: multi-label cross-entropy term in Equation 4-(a) and the zero-bounded term in Equation 4-(b) (Wang et al., 2022b). In Equation 4-(a), all positive logits should surpass all negative logits. In Equation 4-(b), positive logits are anticipated to be greater than 0, and negative logits should be less than 0. As a result, all unit bounds are restricted to 0, which may impose a stringent constraint on specific labels.

Similarly, Equation 5 is derived from H2B loss in Equation 3.

$$\mathcal{L}_{\text{H2B}}^{\mathcal{U}} = \log(1 + \underbrace{\sum_{p \in \mathcal{N}_{\text{pos}}^{\mathcal{U}}} \sum_{n \in \mathcal{N}_{\text{neg}}^{\mathcal{U}}} e^{l_n^{\mathcal{U}} - l_p^{\mathcal{U}} + \left(b_{\text{pos}}^{\mathcal{U}} + b_{\text{neg}}^{\mathcal{U}}\right)}}_{(a)} + \underbrace{\sum_{p \in \mathcal{N}_{\text{pos}}^{\mathcal{U}}} e^{\left(t^{\mathcal{U}} + b_{\text{pos}}^{\mathcal{U}}\right) - l_p^{\mathcal{U}}} + \sum_{n \in \mathcal{N}_{\text{neg}}^{\mathcal{U}}} e^{l_n^{\mathcal{U}} - \left(t^{\mathcal{U}} - b_{\text{neg}}^{\mathcal{U}}\right)}}_{(b)}) \quad (5)$$

H2B loss consists of two components. Equation 5-(a) represents the marginal multi-label cross-entropy, where the margin is defined as the sum of positive and negative biases. When a unit's biases increase, the loss is computed to enhance the differentiation between positive and negative logits, thereby emphasizing the unit's significance. In Equation 5-(b), the bounds for a unit are determined to address the issue of static thresholding, which uses a fixed bound set to 0. The bias of a positive label increases the corresponding bound, whereas the bias of a negative label decreases the corresponding one. As the biases of a unit increase, the loss encourages the logits to create a more distinct separation between biased bounds, as in the left of Figure 2-(c). Conversely, when the biases of a unit decrease to zero, the loss focuses on estimating the bounds between positive and negative logits, as in the right of Figure 2-(c).

**Implementations on Unit-based Models**    To validate the effectiveness of the H2B loss, we have applied it to two recent unit-based HTC models, HPT (Wang et al., 2022b) and HiDEC (Im et al., 2023). These models employ distinct strategies for partitioning a hierarchy into a set of units. In HPT, the same units are utilized during both training and inference. In contrast, HiDEC exhibits variability in its units. This difference stems from the fact that in HiDEC, for a document $x_d$, units are constructed using the target label set $\mathcal{Y}_d$ during training, whereas during inference, units are formed through sub-hierarchy expansion starting from the root. Specifically, in HPT, each unit encompasses all labels at the same hierarchy level. We denote a unit and a target label set for the $m$-th level as $\mathcal{U}_m = \{v_i | \text{level}(v_i) = m, v_i \in \mathcal{V}\}$ and $\mathcal{Y}_d^{\mathcal{U}_m} = \{v_i | v_i \in \mathcal{Y}_d \cap \mathcal{U}_m\}$, respectively. In HiDEC, for a given document $x_d$, a sub-hierarchy label set $\mathcal{V}_d = \mathcal{Y}_d \cup \{v_i | v_i \in \text{ancestor}(v_j), v_j \in \mathcal{Y}_d\}$ and a sub-hierarchy sequence $\text{H}^d = [v_i | v_i \in \mathcal{V}_d \backslash \text{leaf}(\mathcal{G})]$ are created sequentially. Based on $\text{H}^d$, the $k$-th parent unit is defined as $\mathcal{U}_k = \{v_i | v_i \in \text{child}(\text{H}_k^d)\} \cup \{v_{\text{end}}\}$, where $v_{\text{end}}$ is a special node used to terminate sub-hierarchy expansion. Then, a target label set is defined as $\mathcal{Y}_d^{\mathcal{U}_k} = \{v_i | v_i \in \mathcal{V}_d \cap \mathcal{U}_k\}$. For a label assignment, we re-define $\mathcal{Y}_d^{\mathcal{U}_k} = \mathcal{Y}_d^{\mathcal{U}_k} \cup \{v_{\text{end}}\}$ if $\text{H}_k^d \in \mathcal{Y}_d$. In both HPT and HiDEC, a simple feed-forward network (FFN) is employed to learn optimal bounds based on unit representations. Consequently, HPT and HiDEC using H2B loss require only a modest number of additional parameters compared to the original models.

## 4    EXPERIMENTS

### 4.1    EXPERIMENTAL SETTINGS

Table 1: Data statistics. Level and $|\mathcal{V}|$ are the maximum level and number of labels in a hierarchy, while $|\mathcal{W}|$ is the number of units. $|\mathcal{Y}_d|$ and $|\mathcal{W}_d|$ are the average number of target labels and units for a document, while $|\mathcal{U}|$ is the average number of labels in a unit. $|\mathcal{N}_{\text{pos}}^{\mathcal{U}}|$ and $|\mathcal{N}_{\text{neg}}^{\mathcal{U}}|$ are the average number of positive and negative labels for units, respectively. Note that values partitioned by '/' indicate HPT and HiDEC in order.

| Dataset | $|\mathcal{V}|$ | $|\mathcal{W}|$ | Level | Average of | | | | | Train | Dev | Test |
|---|---|---|---|---|---|---|---|---|---|---|---|
| | | | | $|\mathcal{Y}_d|$ | $|\mathcal{W}_d|$ | $|\mathcal{U}|$ | $|\mathcal{N}_{\text{pos}}^{\mathcal{U}}|$ | $|\mathcal{N}_{\text{neg}}^{\mathcal{U}}|$ | | | |
| RCV1-v2 | 103 | 4/22 | 4 | 3.24 | 4/2.98 | 25.75/5.63 | 0.80/1.77 | 24.95/3.86 | 20,833 | 2,316 | 781,265 |
| NYT | 166 | 8/52 | 8 | 7.60 | 8/6.94 | 20.75/4.17 | 0.95/1.79 | 19.80/2.38 | 23,345 | 5,834 | 7,292 |
| EURLEX57K | 4,271 | 6/1,168 | 6 | 5.00 | 6/9.16 | 752.17/5.15 | 0.85/1.06 | 751.32/4.09 | 45,000 | 6,000 | 6,000 |

**Datasets and Evaluation Metrics**    We selected two small-scale datasets, RCV1-v2 (Lewis et al., 2004) and NYT (Sandhaus, 2008), and a large-scale dataset, EURLEX57K (Chalkidis et al., 2019), for our standard experiments. To ensure a fair comparison, we adhered to the same data configuration as previous research (Zhou et al., 2020; Chen et al., 2021; Wang et al., 2022b; Im et al., 2023) and used Micro-F1 and Macro-F1 as our evaluation metrics. Table 1 presents the data statistics for three datasets. RCV1-v2 offers limited training data, while EURLEX57K provides a large number of labels. It is particularly noteworthy to examine the statistics of units. HPT (Wang et al., 2022b) generates a considerably smaller number of units compared to HiDEC (Im et al., 2023). We can see label imbalance explicitly as both HPT and HiDEC produce a limited number of positive but substantial negative labels. As a hierarchy size increases, label imbalance becomes pronounced in HPT, while it remains stable in HiDEC.

**Implementation Details**    We implemented H2B, BCE, and ZMLCE losses using the original codes[1] based on HPT and HiDEC. The same model architectures and hyperparameters were utilized for three datasets.

In HPT, `bert-base-uncased` (Devlin et al., 2019) and GAT (Veličković et al., 2018) were used as text and structure encoders, respectively. The batch size was set to 16. Adam (Kingma & Ba, 2015) optimizer was used with a learning rate of 3e-5. The early stop was applied when Macro-F1 for developments set after each epoch did not increase during 6 epochs. The other hyperparameters were not tuned.

---

[1]Check out code repositories referred to in HPT and HiDEC papers.

Table 2: Overall performance. The upper shows the official scores reported in the original papers, whereas the lower presents the scores from our implementations, with each score accompanied by its standard deviation. Values are derived by averaging results from ten runs with random weight initialization. * indicates that an auxiliary loss is used with the classification loss, while ‿ represents the baseline loss for each model. † and ‡ denotes Wang et al. (2022a) and Chalkidis et al. (2019), respectively.

| Model | Approach | Loss | RCV1-v2 | | NYT | | EURLEX57K | |
|---|---|---|---|---|---|---|---|---|
| | | | Micro-F1 | Macro-F1 | Micro-F1 | Macro-F1 | Micro-F1 | Macro-F1 |
| BERT†‡ | Global | BCE | $85.65^\dagger$ | $67.02^\dagger$ | $78.24^\dagger$ | $66.08^\dagger$ | $73.20^\ddagger$ | - |
| HiAGM (Zhou et al., 2020) | Global | BCE* | 85.58 | 67.35 | 78.64 | 66.76 | - | - |
| HiMatch (Chen et al., 2021) | Global | BCE* | 86.33 | 68.66 | - | - | - | - |
| HGCLR (Wang et al., 2022a) | Global | BCE* | 86.49 | 68.31 | 78.86 | 67.96 | - | - |
| HiTIN (Zhu et al., 2023) | Global | BCE* | 86.71 | 69.95 | 79.65 | 69.31 | - | - |
| HPT (Wang et al., 2022b) | Unit | ZMLCE* | 87.26 | 69.53 | **80.42** | **70.42** | - | - |
| HiDEC (Im et al., 2023) | Unit | BCE | **87.96** | **69.97** | 79.99 | 69.64 | 75.29 | - |
| **Our Implementations** | | | | | | | | |
| HPT | Unit | BCE* | $87.65_{\pm0.11}$ | $69.87_{\pm0.40}$ | $79.49_{\pm0.22}$ | $68.66_{\pm0.30}$ | $71.57_{\pm0.58}$ | $25.34_{\pm0.59}$ |
| | | ZMLCE* | $\mathbf{87.82}_{\pm0.14}$ | $\mathbf{70.23}_{\pm0.31}$ | $80.04_{\pm0.23}$ | $69.69_{\pm0.49}$ | $75.54_{\pm0.20}$ | $\mathbf{28.46}_{\pm0.26}$ |
| | | H2B* | $87.74_{\pm0.11}$ | $69.31_{\pm0.94}$ | $\mathbf{80.23}_{\pm0.17}$ | $\mathbf{70.07}_{\pm0.52}$ | $\mathbf{76.17}_{\pm0.04}$ | $28.33_{\pm0.22}$ |
| HiDEC | Unit | BCE | $\mathbf{87.70}_{\pm0.12}$ | $70.82_{\pm0.20}$ | $80.13_{\pm0.16}$ | $69.80_{\pm0.24}$ | $75.14_{\pm0.19}$ | $27.91_{\pm0.11}$ |
| | | ZMLCE | $87.59_{\pm0.18}$ | $70.61_{\pm0.36}$ | $80.25_{\pm0.21}$ | $70.14_{\pm0.23}$ | $76.17_{\pm0.13}$ | $\mathbf{28.73}_{\pm0.22}$ |
| | | H2B | $87.65_{\pm0.12}$ | $\mathbf{71.52}_{\pm0.31}$ | $\mathbf{80.35}_{\pm0.12}$ | $\mathbf{70.59}_{\pm0.32}$ | $\mathbf{76.43}_{\pm0.08}$ | $28.60_{\pm0.18}$ |

In HiDEC, `bert-base-uncased` was used as a text encoder, while a 2-layer transformer decoder (Vaswani et al., 2017) was used as a hierarchy decoder. The label embeddings were initialized using a normal distribution with $\mu = 0$ and $\sigma = 768^{-0.5}$. The batch size was set to 64. AdamW (Loshchilov & Hutter, 2019) optimizer was used with the learning rate 5e-5. The learning rate was scheduled using a linear scheduler with a warmup rate of 0.1 over 100 epochs.

**Comparison Models** For comparison, we selected recent HTC models that leverage pre-trained language models: HiAGM (Zhou et al., 2020), HiMatch (Chen et al., 2021), HGCLR (Wang et al., 2022a), HiTIN (Zhu et al., 2023), HPT (Wang et al., 2022b), and HiDEC (Im et al., 2023).

**HiAGM:** HiAGM utilizes the prior probability of parent-child label dependency as adjacency of Graph Convolution Networks (GCN) (Kipf & Welling, 2017). A text representation from a text encoder such as TextRCNN (Lai et al., 2015) or BERT is propagated to GCN using text propagation.

**HiMatch:** HiMatch considers HTC as a semantic matching problem and conducts text and label semantic matching to HiAGM through a hierarchy-aware matching loss. In addition, the hierarchy-aware margin loss learns to adjust the distance based on the label's hierarchical relationship to reflect hierarchy in presentation.

**HGCLR:** HGCLR points out the limitations of the existing models that use separate text and structure encoders and proposes a contrastive learning method that can inject structural information into the text encoder.

**HiTIN:** To address the limitations of the existing global approach, HiTIN employs a strategy of reconstructing the hierarchy into a code tree to reduce structural complexity effectively. This code tree construction aims to minimize structural entropy, resulting in a simplified hierarchy that maximizes the retention of structural information from the original hierarchy.

**HPT:** HPT is the first attempt to address HTC using prompt tuning. It transforms HTC into a hierarchy-aware multi-label MLM to incorporate the HTC and MLM. The hierarchy representation at different levels, represented through GAT, is used in conjunction with text as input to BERT. Classification is performed for labels corresponding to units at each level.

**HiDEC:** To address the issue of excessive parameters in the existing models, HiDEC employs a sub-hierarchy composed of labels related to documents rather than the entire hierarchy. HiDEC transforms HTC into a sequence generation problem and conducts training to generate sub-hierarchy sequences.

HiAGM, HiMatch, HGCLR, and HiTIN are global models, whereas HPT and HiDEC are unit-based models. All models employ BERT as a text encoder. Apart from HPT utilizing ZMLCE loss, the other models use BCE loss.

## 4.2 RESULTS

Table 2 presents the overall performance of three datasets. The scores and their variances were obtained from our implementations by averaging results from ten runs with random weight initialization. On NYT, H2B improved over the baseline losses, ZMLCE and BCE, in both HPT and HiDEC. In HiDEC, the performance exhibited an ascending trend with BCE, ZMLCE, and H2B in order. However, a different pattern emerged on EURLEX57K. Compared to H2B, BCE significantly deteriorated the performance, highlighting the need to address label imbalance in such a large-scale hierarchy. Interestingly, ZMLCE and H2B in HPT and HiDEC resulted in a gain in Micro-F1, which led to a drop in Macro-F1. Bounds and biases in H2B seem to impact unit performance and lead to a tradeoff. On RCV1v2, H2B loss resulted in only minor degradation, except for HiDEC in Macro-F1. These results came from the variance used in positive and negative biases, as detailed in Section 4.3, along with the analysis of other results.

## 4.3 ABLATION STUDIES

Table 3: Ablation results of H2B on three datasets. The first ranks are highlighted in red-bold in each group, while the second ones are underlined.

| Model | | Bounds | Biases | RCV1v2 | | NYT | | EURLEX57K | |
|---|---|---|---|---|---|---|---|---|---|
| | | | | Micro F1 | Macro F1 | Micro F1 | Macro F1 | Micro F1 | Macro F1 |
| HPT | (a) | - | - | 87.82 | 70.23 | 80.04 | 69.69 | 75.54 | **28.46** |
| | (b) | ○ | - | **87.82** | **70.37** | 80.01 | 69.71 | 75.58 | 28.37 |
| | (c) | - | ○ | 87.69 | 69.29 | 80.19 | 69.90 | 76.17 | 28.40 |
| | (d) | ○ | ○ | 87.74 | 69.31 | **80.23** | **70.07** | 76.17 | 28.33 |
| HiDEC | (a) | - | - | 87.59 | 70.61 | 80.25 | 70.14 | 76.17 | **28.73** |
| | (b) | ○ | - | **87.74** | 70.87 | 80.29 | 70.15 | 76.11 | 28.66 |
| | (c) | - | ○ | 87.43 | 71.01 | **80.38** | 70.51 | 76.35 | 28.58 |
| | (d) | ○ | ○ | 87.65 | **71.52** | 80.35 | **70.59** | 76.43 | 28.60 |

We conducted ablation studies to analyze the impact of bounds and biases in H2B. As shown in Equations 2 and 3, H2B becomes equivalent to ZMLCE when the bounds are set to 0 and biases are removed. The results of these ablation studies are presented in Table 3.

On NYT and EURLEX57K, bounds have little impact on ZMLCE, while biases consistently lead to improvements. However, on EURLEX57K, bounds and biases lead to a slight decrease in Macro-F1. The reason is that severe label imbalance, 79% of labels appearing fewer than 50 times, causes unstable estimation of bounds and large biases. On RCV1-v2, bounds contribute to improvement, while biases lead to a decrease in Micro-F1. Specifically, bounds result in an improvement over ZMLCE. However, biases decrease Micro-F1 in both HPT and HiDEC, as well as Macro-F1 in HPT. H2B negatively affects performance in HPT but shows improvement, particularly in Macro-F1, in HiDEC. This difference arises because most few-shot labels are concentrated within two units in HPT, whereas they spread across units in HiDEC. A detailed analysis will be provided in Subsection 4.4.

## 4.4 ANALYSIS OF H2B BASED ON LABEL FREQUENCY

For an in-depth investigation of label imbalance, we partitioned a dataset into frequent, few-shot, and zero-shot label sets based on label frequencies like EURLEX57K (Chalkidis et al., 2019). The frequent refers to labels with occurrences exceeding a predefined cutoff threshold, while the few-shot encompasses the remaining labels that appeared in training. The zero-shot refers to labels that never appeared in training but were excluded from the analysis since both models do not consider them. The cutoffs were set at 50 for three datasets. Figure 3 illustrates the Macro-F1 gains in Table 3. Based on the blue vertical line, the left and right denote the frequent and few-shot labels, respectively. The labels are sorted in descending order based on label frequencies. Subsequently, the gain is defined as the difference in Macro-F1, computed up to the corresponding label, between H2B ablation (one of b, c, d) and ZMLCE (a).

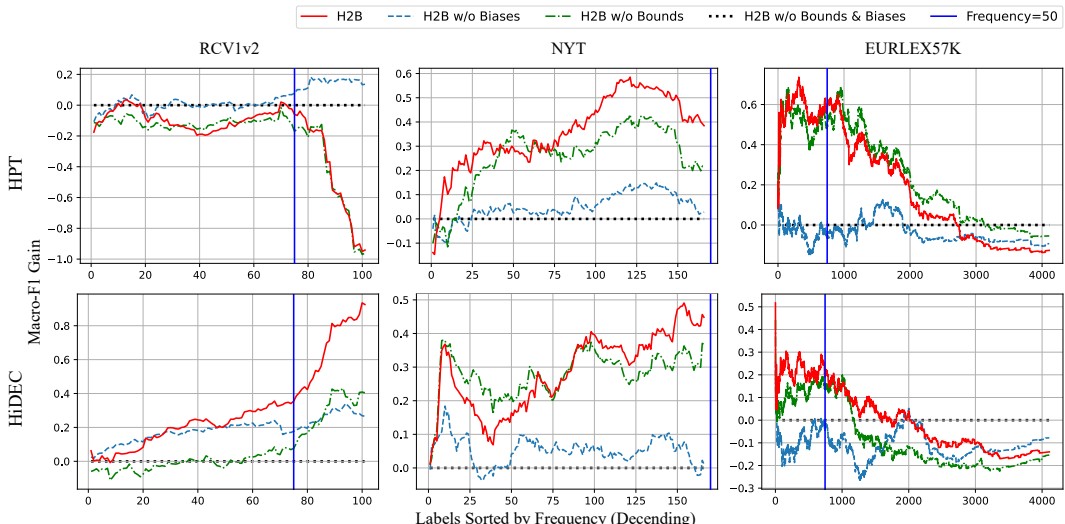

Figure 3: Macro-F1 gains in Table 3. A gain is defined as (ablation setting – ZMLCE) over the F1 score for each label. The gains are sorted in descending order using label frequency. The graph illustrates Macro-F1 gain up to a corresponding label. Based on the blue vertical bar, the left is the frequent labels, while the right is the few-shot labels.

As depicted in Figure 3, H2B exhibits consistent patterns in HPT and HiDEC across NYT and EU-RLEX57K. H2B enhances the performance of frequent labels while diminishing the performance of few-shot labels. This underscores the adverse effect of applying H2B to datasets with limited training data. However, in contrast to NYT and EURLEX57K, the influence of H2B on few-shot labels depends on the model used on RCV1-v2. The macro F1 score for few-shot labels decreases in HPT but increases in HiDEC. This discrepancy arises because the two models' distinct unit configurations lead to dissimilar distributions over labels, particularly few-shot labels. While HiDEC distributes 27 few-shot labels across 11 units, HPT concentrates its label configuration within only 2 units. This imbalance results in unstable logits and significantly impacts H2B, which relies on the logit's standard deviation as a measure of bias. In summary, H2B consistently improves the performance of frequent labels but exhibits instability when applied to few-shot labels. However, the comparison between HPT and HiDEC on RCV1-v2 highlights that substantial improvements can be achieved depending on the structural configuration of units.

## 4.5    ANALYSIS OF BOUNDS AND BIASES

Figure 4 illustrates the samples of thresholds, biased bounds, and logits obtained from ZMLCE and H2B during inference in HPT. Each point on the graph represents a logit, with target labels in green and non-target labels in red, respectively. The logits are obtained from test documents in NYT (a and b) and EURLEX57K (c) using HPT's level 1 and 2 units. Blue lines denote the threshold used in each unit. In H2B, a threshold is determined by a bound predicted for each unit, based on a specific document. The green and red lines indicate positive and negative biased bounds of units, respectively, in H2B. They are not used during inference.

As in Figure 4, H2B employs dynamic thresholds for each document and unit, whereas ZMLCE applies a zero threshold to all units. The dynamic thresholds reduce false predictions, as in Figure 4-(a) and (c). As biases influence the distinction between positive and negative labels, the biased bounds (green and red lines) lead to more significant deviations in positive and negative logits than the unbiased bound (blue line). They enhance the discriminative power of a model by encouraging the logits to move further away from the threshold, as in Figure 4-(b). As depicted in Figure 4-(c), we observed higher thresholds when a unit comprises many few-shot labels, such as EURLEX57K. This phenomenon arises because raising the bounds is relatively easier than lowering all negative logits associated with few-shot labels in training. Consequently, attaining logits higher than the bound

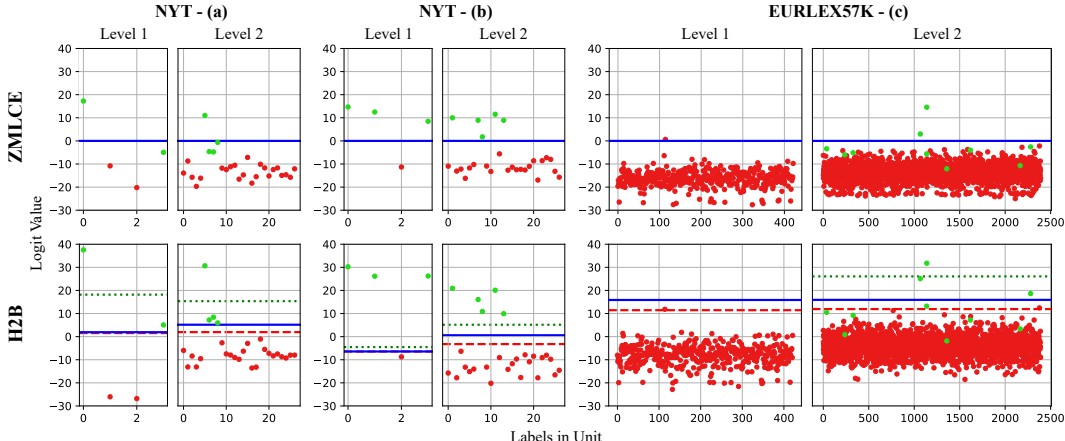

Figure 4: Illustration of sample logits of documents obtained from ZMLCE and H2B during inference (a, b and c) in HPT. Each point on the graph represents a logit, with target labels in green and non-target labels in red, respectively. Blue lines denote the threshold used in each unit, while the green and red lines indicate positive and negative biased bounds, respectively, in H2B. In (a), H2B effectively reduces false predictions through dynamic thresholding. In (b), logits obtained with H2B are clearly distinguishable beyond the biased bounds. In (c), higher thresholds are observed when a unit comprises many few-shot labels.

for the few-shot positive labels is difficult. This issue can be mitigated by reducing the number of few-shot labels, as observed on NYT, or adjusting the unit configuration, as exemplified in HiDEC.

## 5 CONCLUSION

This paper introduces a Hierarchy-aware Biased Bound (H2B) loss function, offering two key innovations to address the challenges of static thresholding and label imbalance in HTC. First, H2B introduces learnable bounds for all units within a hierarchy to address static thresholding. These bounds are optimized for various units during training and are used as dynamic thresholds during inference. Second, H2B introduces biases for both positive and negative labels to alleviate the label imbalance, where a bias measures the degree of imbalance within a unit. The biases help to adjust the bounds to reduce the impact of the ratio between positive and negative labels. Extensive experiments on benchmark HTC datasets demonstrate the superiority of H2B loss based on unit-based HTC models by comparing competitive HTC models and comprehensive analysis. We plan to extend H2B to extremely large-scale hierarchies and improve imbalance relations among units.

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

## A    DATASET DETAILS

We provide a more detailed examination of the datasets as presented in Table 1, yielding several key observations:

- **Train-Test Mismatch:** In RCV1-v2, there is a notable disparity in the sizes of the training and test sets, leading to a train-test mismatch.

- **Label Hierarchy Disparities:** EURLEX57K has a label hierarchy of 42 times and 25 times larger than RCV1-v2 and NYT, respectively. This substantial discrepancy in size causes a significant imbalance between positive and negative labels. The average of $|\mathcal{N}_{\text{pos}}^{\mathcal{U}}|$ remains relatively stable, while the average of $|\mathcal{N}_{\text{neg}}^{\mathcal{U}}|$ increases significantly from 24.95 and 19.80 in RCV1-v2 and NYT to 751.32 in EURELX57K.

- **Unit Imbalance:** The disparity in the unit construction strategies between HPT and HiDEC leads to substantial variations in unit statistics. HiDEC divides the hierarchy into smaller units than HPT, resulting in a small number of labels for each unit (Average of $|\mathcal{U}|$) and significantly balances the ratio of positive and negative labels for each unit (Average of $|\mathcal{N}_{\text{pos}}^{\mathcal{U}}|$ and $|\mathcal{N}_{\text{neg}}^{\mathcal{U}}|$). However, HiDEC still suffers from label imbalance.

Additionally, EURLEX57K is categorized into three types based on label frequencies: "frequent" labels are those that appeared more than 50 times in the training data, "few-shot" labels are those that appeared less than 50 times, and "zero-shot" labels are those that have never appeared. This paper focuses on frequent and few-shot labels, as our baseline models, HPT and HiDEC, were not designed to handle zero-shot settings.

## B    ABLATION RESULT BY FREQUENCY

Table 4 presents a comprehensive ablation study for frequent, few-shot, and overall labels using H2B. All performances are averaged from ten runs with random weight initialization. On RCV1v2, biases negatively affect both frequent and few-shot labels in HPT, leading to a decline in performance. On the other hand, HiDEC shows an improvement in Macro-F1 for frequent labels and few-shot labels. On EURLEX57K, biases improve both Micro-F1 and Macro-F1 scores for frequent labels. However, for few-shot labels, while Micro-F1 increases, Macro-F1 decreases due to the impact of biases.

Table 4: Ablation results of H2B on three datasets, including frequent and few-shot performances. The number in parentheses indicates the number of labels according to frequency. MiF and MaF denotes Micro-F1 and Macro-F1, respectively.

| Model | | Bounds | Biases | RCV1v2 | | | | | | NYT | | EURLEX57K | | | | | | | |
|---|---|---|---|---|---|---|---|---|---|---|---|---|---|---|---|---|---|---|---|
| | | | | Frequent (76) | | Few-shot (27) | | Overall (103) | | Overall (166) | | Frequent (746) | | Few-shot (3,362) | | Overall (4,271) | | | |
| | | | | MiF | MaF | MiF | MaF | MiF | MaF | MiF | MaF | MiF | MaF | MiF | MaF | MiF | MaF | | |
| HPT | (a) | - | - | 88.18 | 78.76 | 58.39 | 46.22 | 87.82 | 70.23 | 80.04 | 69.69 | 78.58 | 69.84 | 54.88 | **20.65** | 75.54 | **28.46** | | |
| | (b) | ○ | - | **88.18** | **78.87** | **58.98** | **46.45** | **87.82** | **70.37** | 80.01 | 69.71 | 78.62 | 69.77 | 55.03 | 20.55 | 75.58 | 28.37 | | |
| | (c) | - | ○ | 88.05 | 78.61 | 56.24 | 43.06 | 87.69 | 69.29 | 80.19 | 69.90 | 79.17 | 70.33 | **55.47** | 20.48 | 76.17 | 28.40 | | |
| | (d) | ○ | ○ | 88.11 | 78.70 | 56.02 | 42.89 | 87.74 | 69.31 | **80.23** | **70.07** | 79.19 | **70.41** | 55.33 | 20.37 | **76.17** | 28.33 | | |
| HiDEC | (a) | - | - | 87.94 | 78.69 | 59.05 | 47.88 | 87.59 | 70.61 | 80.25 | 70.14 | 79.27 | 70.86 | 55.17 | **20.78** | 76.17 | **28.73** | | |
| | (b) | ○ | - | **88.08** | 78.87 | 59.74 | 48.35 | **87.74** | 70.87 | 80.29 | 70.15 | 79.22 | 70.83 | 54.99 | 20.69 | 76.11 | 28.66 | | |
| | (c) | - | ○ | 87.78 | 78.79 | 60.60 | 49.09 | 87.43 | 71.01 | **80.38** | 70.51 | 79.40 | 71.01 | **55.23** | 20.56 | 76.35 | 28.58 | | |
| | (d) | ○ | ○ | 87.98 | **79.07** | **61.32** | **50.25** | 87.65 | **71.52** | 80.35 | **70.59** | 79.50 | **71.05** | 55.23 | 20.56 | **76.43** | 28.60 | | |

## C  MORE SAMPLES OF UNIT PREDICTIONS

In Figure 5, we display the complete unit predictions for the sample presented in Figure 4, with HPT.

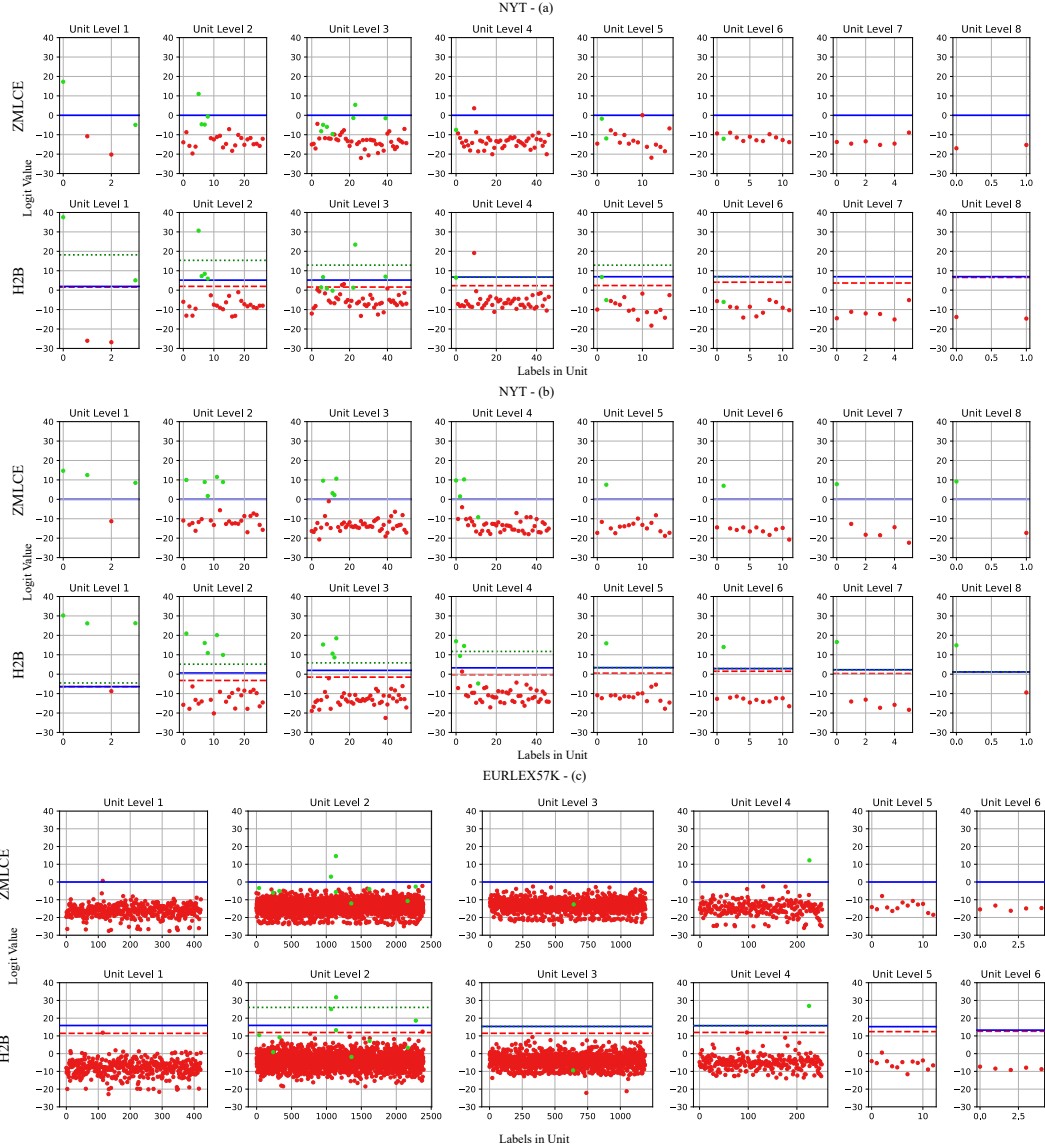

Figure 5: Complete unit predictions of samples in Figure 4, with HPT.

