# OpenReview forum: "Hierarchy-aware Biased Bound Loss Function for Hierarchical Text Classification"
_ICLR.cc/2024/Conference — ICLR 2024 Conference Withdrawn Submission_

### Official Review · Reviewer_LxBt · 2023-10-16

**Soundness:** 3 good
**Presentation:** 3 good
**Contribution:** 2 fair
**Rating:** 5
**Confidence:** 4

**Summary:**

This paper proposes the Hierarchy-aware Biased Bound (H2B), aiming to reduce the bias and solve the imbalance problem in the hierarchical text classification task. The authors achieve the above goal by introducing learnable bounds and biases to further push the logits of the positive label set to become more positive and the logits of the negative label set to become more negative. Experimenting on three datasets by comparing with six baselines and two different losses, the authors' proposed method achieves competitive results.

**Strengths:**

1. The proposed method is easy to understand.
2. This paper is well-organized, with clear experiment settings and results. The authors also provide comparison and ablation results in the figure for better understanding.

**Weaknesses:**

The soundness of this paper is good, but there exist some critical problems with the proposed method. The authors seem to misunderstand the ZMLCE loss. According to `Wang et al., 2022`, the definition of ZMLCE is
$$
\mathcal{L}_{\text{ZMLCE}} = \frac{1}{|\mathcal{W}|} \sum _{\mathcal{U}\in\mathcal{W}} \left( \log \left( 1+\sum _{ p\in\mathcal{N} _{\text{pos}}^{\mathcal{U}} } e^{-l_p^{\mathcal{U}} + b_p^{\mathcal{U}} } \right) + \log \left( 1+\sum _{ p\in\mathcal{N} _{\text{neg}}^{\mathcal{U}} } e^{l_n^{\mathcal{U}} + b_n^{\mathcal{U}} } \right) \right),
$$
which contains the learnable bias in the experiential term. The authors' proposed method is to add a learnable bound $t _{\mathcal{U}}$ based on the above formulation. This will not work because the function of $t _{\mathcal{U}}$ is overlapped with the learnable bias $b_p^{\mathcal{U}}$ and $b_n^{\mathcal{U}}$. As a result, the $t _{\mathcal{U}}$ becomes an **amplifier** but does not serve as a "bound", and the experiment results in Table 2 and Figure 4 also demonstrate the above concern. I think the author should reconsider the proposed method.

Wang et al. HPT: Hierarchy-aware prompt tuning for hierarchical text classification. EMNLP 2022

**Questions:**

N/A

---

### Official Review · Reviewer_CjA1 · 2023-10-30

**Soundness:** 2 fair
**Presentation:** 3 good
**Contribution:** 2 fair
**Rating:** 3
**Confidence:** 3

**Summary:**

The paper introduces the H2B (Hierarchy-aware Biased Bound) loss to enhance Hierarchical Text Classification (HTC). H2B aims to improve the accuracy of category assignments by adapting the decision boundaries for each category. HTC involves classifying text into a predefined label hierarchy. Since some categories are more specific than others, there is an inherent class imbalance, i.e., there may be a lot of examples for some categories but very few for others. H2B addresses this by allowing the model to learn and adjust its decision boundaries for each category in the hierarchy. H2B also introduces weight adjustments for positive and negative labels, to guide the model to pay more attention to underrepresented labels, helping it better learn and distinguish between categories with imbalanced examples.

**Strengths:**

-  To the best of my knowledge, and after reviewing several loss functions in the literature, H2B seems a uniquely designed loss that combines bias and hierarchy-aware components into a single loss function, tailored to address the challenges specific to hierarchical text classification.
- The experiments were conducted over several global and recent unit-based HTC models, HPT and HiDEC, on three benchmark datasets.

**Weaknesses:**

- The paper reports that H2B has mixed results on few-shot labels, with a decrease in Micro-F1 for HPT and an increase in HiDEC on some datasets. This mixed performance on less common labels suggests that H2B may not be universally beneficial and could be sensitive to the specific characteristics of the dataset and model configuration. The paper does not provide a clear explanation of why such variations occur.
- Performance improvements over existing methods in Table 2 are marginal. The paper also mentions that severe label imbalance causes
unstable estimation of bounds and large biases, and moreover reports that H2B has mixed results on few-shot labels, with a decrease in Micro-F1 for HPT and an increase in HiDEC on some datasets. This mixed performance suggests that H2B may not be universally beneficial and could be sensitive to the specific characteristics of the dataset, label hierarchy, and model configuration.
- The paper briefly mentions that label imbalance is a concern in large-scale hierarchies. It would be beneficial to provide a more in-depth analysis of the specific challenges posed by large-scale hierarchies and how H2B helps mitigate these challenges. Current experiments do not seem to support this claim. Perhaps the authors can further explain this.
- The paper presents two sets of results: one from the official scores reported in the original papers and another from the authors' implementations. However, the reasoning for this is unclear and can raise concerns about the reliability and comparability of the findings. Is it because the code for these methods is not available online or where there challenges in replicating the original results?
- It is worth noting in the limitations that H2B assumes that the hierarchical label structure is known and correctly specified. If the hierarchy is not well-defined or is subject to change, this approach may not be suitable. The performance of H2B can also be sensitive to the depth and structure of the label hierarchy.
- An additional suggestion would be to also include qualitative results that show how the proposed method compares to existing approaches (both success and failure cases).

Overall, this work introduces a novel loss function for HTC and presents extensive results over various existing baselines. The paper is *commendable* for its simple and intuitive approach. While the work has potential, the paper's contribution remains unclear. The methodological inconsistency (reporting two sets of results for baselines) and the mixed results in empirical evaluations do not seem sufficient to draw meaningful conclusions and/or effectively advance the state-of-the-art in HTC.

**Questions:**

- Please provide a reference for the claim "The unit-based approach has achieved significant improvements over the global approach."
- Using standard deviation can be sensitive to the scale of the biases. Have the authors considered other approaches, e.g., using median absolute deviation which is less sensitive to outliers?

---

### Official Review · Reviewer_Dnhv · 2023-11-04

**Soundness:** 3 good
**Presentation:** 2 fair
**Contribution:** 1 poor
**Rating:** 3
**Confidence:** 4

**Summary:**

With the massive growth of information in the form of text documents on the internet, it becomes important to automatically categorize and understand them. Hierarchical text classification (HTC) aims to foster this by learning to classify a text document onto a pre-defined semantic hierarchy of labels. Main challenges of HTC involve large label scale, complex label hierarchy and imbalance in positive vs negative labels for a document. This paper addresses the latter label imbalance problem by incorporating separate bias thresholds for +ve and -ve classes. This involves a simple and straight-forward linear modification to existing loss choices. Experiments across multiple datasets and different loss functions show minor gains due to this innovation in terms of F1 metrics.

**Strengths:**

* Paper tackles a moderately problem: bias rectification in hierarchical text classification
* Proposed modification is simple and generically applicable to most HTC loss functions proposed earlier

**Weaknesses:**

* Results show minor and inconsistent gains across datasets. As such, the method does not show a lot of promise
* The technical innovation is quite simple. Relative to previous works in HTC, which introduce significant innovations such as global label interactions, label semantic modeling, prompt tuning for HTC etc., the technical contribution of this paper appears minor in comparison.
* The presentation of bias terms could have been better: e.g. it isn't even made clear how the bias terms are learned or whether they are set by heuristics/domain knowledge

**Questions:**

* Are the bias terms learned or are they set by heuristics/domain knowledge?